# Proto-Caps: interpretable medical image classification using prototype learning and privileged information

Luisa Gallée[1,2], Catharina Silvia Lisson[2,3], Timo Ropinski[2,4], Meinrad Beer[2,3,5] and Michael Götz[1,2,3,5]

[1] Experimental Radiology, Ulm University Medical Center, Germany, Ulm, Germany
[2] XAIRAD—Cooperation for Artificial Intelligence in Experimental Radiology, Germany, Ulm, Germany
[3] Department of Diagnostic and Interventional Radiology, Ulm University Medical Center, Germany, Ulm, Germany
[4] Visual Computing Group, Ulm University, Germany, Ulm, Germany
[5] i2SouI—Innovative Imaging in Surgical Oncology Ulm, Ulm University Medical Center, Germany, Ulm, Germany



Corresponding author
Luisa Gallée, luisa.gallee@uni-ulm.de

## ABSTRACT

Explainable artificial intelligence (xAI) is becoming increasingly important as the need for understanding the model's reasoning grows when applying them in high-risk areas. This is especially crucial in the field of medicine, where decision support systems are utilised to make diagnoses or to determine appropriate therapies. Here it is essential to provide intuitive and comprehensive explanations to evaluate the system's correctness. To meet this need, we have developed Proto-Caps, an intrinsically explainable model for image classification. It explains its decisions by providing visual prototypes that resemble specific appearance features. These characteristics are predefined by humans, which on the one hand makes them understandable and on the other hand leads to the model basing its decision on the same features as the human expert. On two public datasets, this method shows better performance compared to existing explainable approaches, despite the additive explainability modality through the visual prototypes. In addition to the performance evaluations, we conducted an analysis of truthfulness by examining the joint information between the target prediction and its explanation output. This was done in order to ensure that the explanation actually reasons the target classification. Through extensive hyperparameter studies, we also found optimal model settings, providing a starting point for further research. Our work emphasises the prospects of combining xAI approaches for greater explainability and demonstrates that incorporating explainability does not necessarily lead to a loss of performance.

# INTRODUCTION

Deep learning has emerged as a powerful tool in computer vision, excelling in tasks such as object detection, segmentation, and classification. Deep neural networks have the potential to be useful in various domains, including economics, banking, automotive driving, and medicine. They can be employed for decision-making or automation, thereby facilitating

and improving processes (*Secinaro et al., 2021*). When artificial intelligence systems are used in high-risk areas such as medicine, they need to be carefully evaluated to ensure their safety and to gain trust of users (*Rudin, 2019*; *Davenport & Kalakota, 2019*). As the variety of input data cannot be fully covered by test datasets, accuracy tests can only provide limited assurance (*Liang et al., 2022*; *Daneshjou et al., 2021*).

Particularly in medical applications, there can be significant variations in the characteristics of a pathology or imaging, such as differences in equipment between manufacturers (*Cohen et al., 2020*; *Pooch, Ballester & Barros, 2020*). An alternative way of validation is to disclose the reasoning criteria of the artificial intelligence (AI) model (*Leichtmann et al., 2023*). If an AI model justifies its decision in a similar way to a physician as human expert, it appears trustworthy (*Gallée et al., 2024*). However, deep neural networks, also known as black boxes, contain complex data processing steps that are difficult to interpret (*Castelvecchi, 2016*; *Rudin, 2019*; *Gallée et al., 2023*). This difficulty has led to the development of the research field of Explainable AI, which encompasses the development of methods to represent the internal decision-making processes of deep learning models in a way that is understandable to humans.

One approach to model interpretability is the use of *post-hoc* methods, which are applied after training of the primary models. These can be either model-agnostic procedures, which examine the model independently of its architecture, or model-specific techniques that leverage the model's unique characteristics (*van der Velden et al., 2022*; *Murdoch et al., 2019*). While flexible and applicable across various models, *post-hoc* methods may not fully capture the model's internal decision-making process, and the generated explanations may not align with the model's true behavior. Alternatively, intrinsically explainable deep neural networks integrate explanations directly into the model, eliminating the need for *post-hoc* analysis (*Chen et al., 2019*; *Gallée et al., 2023*). While these models provide more transparent insights, they can suffer from trade-offs, such as increased model complexity. Thus, while *post-hoc* methods offer flexibility and broader applicability, intrinsically explainable models tend to provide more accurate, aligned explanations at the cost of other trade-offs. The choice between these approaches depends on the specific needs of the application, including the balance between interpretability and model complexity.

This also applies to the Proto-Caps method (*Gallée, Beer & Götz, 2023*), analyzed in this work. It combines domain knowledge with prototype learning. The diagnostic criteria used by radiologists serve as domain knowledge and are referred to as *attributes*. These attributes are visual characteristics that are essential for classifying a *target*, such as a medical diagnosis. Unlike existing methods that either compute attribute scores or learn target prototypes, Proto-Caps merges both approaches into attribute prototypes with scores. The advantage of these over target prototypes is that the attribute prototypes specifically represent individual attributes and validate the attribute scores.

Technically, the decision-making process is modeled hierarchically, similar to human reasoning. First, a capsule network learns the attributes. These capsules then serve as the core basis for target prediction. In a broader sense, the model first learns visual criteria and

then infers the disease from them. Additionally, the prototypes are derived from the attribute-specific capsules.

This article builds on previous work presented at the 2023 MICCAI Conference (*Gallée, Beer & Götz, 2023*) where we introduced Proto-Caps. The most important aspects of our original contribution include the following:

- We presented a learning algorithm that bases target prediction on attribute-specific prototypes. The prediction is explained with real image examples of these prototypes.
- An evaluation on the LIDC-IDRI dataset for the classification of malignancy of lung nodules showed a higher performance than other explainable methods and a comparable performance to non-explanatory methods.

After demonstrating the methodological feasibility, this article elaborates on the model analysis and provides insights into the information processing of the model. The extensions to the original work can be summarized as follows:

- **Explainability:** We analyze the trustworthiness of the explanations and incorporate feedback from target users.
- **Validation:** We extend the performance evaluation to a second medical dataset and in a three-dimensional environment.
- **Prototype analysis:** We examine the model prototypes for their diversity, which gives an indication of the quality of local explainability.
- **Robustness:** We also analyze the robustness of Proto-Caps with respect to the architectural parameters and provide recommendations for hyperparameters for further research.

## RELATED WORK

This work combines two modalities of explainability to realize the concept of visual prototypes that are understandable to humans. Capsule networks offer an architecture that closely resembles the hierarchical decision-making process of humans. They base the evaluation of an object on learned and specifiable features. On the other hand, we employ a prototype learning strategy to learn feature-specific prototypes that can be used for fine-grained analysis of a model prediction.

### Prototype learning

Prototype learning can enhance model performance across various tasks—including unsupervised domain adaptation, few-shot learning, and reducing catastrophic forgetting in capsule networks (*Pan et al., 2019*; *Sun et al., 2019*; *Snell, Swersky & Zemel, 2017*). Furthermore, example-based approaches have long provided intuitive tools for interpreting neural network decisions, thereby advancing the field of explainable AI (*Bien & Tibshirani, 2011*; *van der Waa et al., 2021*). Global and local explanations of an AI model can be achieved by finding representative examples. Global explanations can be obtained by identifying examples that represent a cluster of a class and best separate the samples

according to their classes. Local explanations can be provided by identifying a prototype that is closest in terms of model features to a sample to be classified (*Barnett et al., 2021*). While global model-explanatory approaches are valuable during model development and for a general overview, local, or case-based explainability provides a sample-specific answer to why the model made its decisions.

By combining explanation by attention learning strategies (*Zhou et al., 2016*; *Zheng et al., 2017*) and prototypes, *Chen et al. (2019)* attracted great attention with prototype-explanation. They introduced a model that provides region-wise prototypes. The specification of the explanations to individual image regions increases their quality, as they are similar to the human intuition of image description. However, despite identifying specific regions, the method does not clarify why the selected prototype is considered similar to that region, leaving an important aspect of interpretability unaddressed.

Also in medical applications, prototypes are often integrated into model architectures to enhance performance (*Wang et al., 2024*; *LaLonde, Torigian & Bagci, 2020*) or improve interpretability (*Hesse & Namburete, 2022*; *Wolf, Pölsterl & Wachinger, 2023*; *Li et al., 2018*). Typically, these methods employ class-specific prototype learning and often generate synthetic images as prototypes. In contrast, our approach uses real samples from the training dataset as prototypes, as in *Chen et al. (2019)*, thereby avoiding unrealistic prototype images. Most notably, our method distinguishes itself by employing attribute-specific prototypes, which enable even finer-grained explanations for complex vision tasks compared to traditional class-specific prototypes.

In addition to the feasibility evaluation, we would like to take up the point made by *Pathak et al. (2024)* in this article and analyze the learned prototypes in more detail. A comprehensive evaluation of prototype networks requires further analysis of the prototypes to examine not only their performance but also their quality. Following this need, we present experimental results on the truthfulness and diversity of the prototypes.

## Privileged information in capsule network

Besides prototype learning, our approach utilizes a second method, which is based on the capsule network architecture in combination with privileged information. Additional domain expertise can enhance the trustworthiness of AI models (*van der Velden et al., 2022*) by providing valuable information about the target objects. In computer vision, it is particularly valuable to integrate and be able to predict the specific visual attributes that an object fulfills for classification (*Shen et al., 2019*).

The capsule learning strategy, originally proposed by *Sabour, Frosst & Hinton (2017)*, represents an attention between encapsulated low and high-level features in a hierarchical manner, similar to that of the human visual system. Capsule networks have been used in various projects and have proven to be a high-performance backbone (*Afshar, Mohammadi & Plataniotis, 2018*).

Our work follows a similar approach to that of *LaLonde, Torigian & Bagci (2020)* and combines additional expert knowledge into capsule networks. They map encapsulated features, which serve as the information base for target classification, to human-defined high-level visual attributes. By predicting these attributes, this creates human

understandable explainability. However, these attribute scores lack validation, which our approach fulfills with attribute-specific visual prototypes.

# METHODS AND MATERIALS

## Datasets

For the evaluation of our method, we used two medical benchmark datasets selected for their detailed annotations, including target classifications and attribute labels. Since our approach predicts the target based on these attributes—its decision criteria—these datasets effectively represent the problem we aim to solve.

### Dataset 1: lung nodule classification

The Lung Image Database Consortium and Image Database Resource Initiative (LIDC-IDRI) dataset (*Armato et al., 2015*) is a comprehensive collection of computed tomography (CT) scans, annotated by expert radiologists, with the primary focus on lung nodule detection and characterization (*Armato et al., 2011*). The annotations contain detailed information on the medical assessment and visual appearance of pulmonary nodules. Each scan was reviewed by up to four radiologists. Identified lung nodules were segmented and annotated with a malignancy rating and visual attributes. The malignancy rating ranges from 1-*highly unlikely* to 5-*highly suspicious*. The visual appearance is described by the attributes subtlety (difficulty of detection, 1-*extremely subtle*, 5-*obvious*), internal structure (1-*soft tissue*, 4-*air*), pattern of calcification (1-*popcorn*, 6-*absent*), sphericity (1-*linear*, 5-*round*), margin (1-*poorly defined*, 5-*sharp*), lobulation (1-*no lobulation*, 5-*marked lobulation*), spiculation (1-*no spiculation*, 5-*marked spiculation*), and texture (1-*non-solid*, 5-*solid*).

Preprocessing for this dataset involves ignoring lung nodules smaller than 3 mm and those identified by less than three radiologists. In the case of 2D processing the lung nodules were cropped out of the volumes with the minimum square bounding box and the slices were resized to $32 \times 32$, as in previous work (*LaLonde, Torigian & Bagci, 2020*). Considering each annotation and slice as a sample, the preprocessing results in a total of 27,379 samples. For 3D processing, a fixed depth was chosen with a centered bounding box, resulting in a volume size of $32 \times 32 \times 16$, according to existing work (*Mehta et al., 2021*). Considering each annotation as a sample, the preprocessing results in a total of 4,318 samples. Preprocessing was performed with the `pylidc` framework (*Hancock & Magnan, 2016*).

Each sample comes with the individual annotator's segmentation, while the target and attribute labels take into account the annotations of all annotators, as in previous work (*LaLonde, Torigian & Bagci, 2020*). For the nodule's attribute score the mean value of the radiologists' scores is used. The target ground truth is represented with the distribution of the annotated malignancy scores.

Experiments with this dataset were performed using five-fold stratified cross-validation, with a patient-wise split and using 10% of the training data for validation.

### Dataset 2: lung abnormality classification

As second dataset to validate the proposed method we chose the CheXpert dataset which contains 224,316 chest radiographs of 65,240 patients (*Irvin et al., 2019*). In this dataset the testing set includes labels which were obtained by expert radiologists, whereas the training labels were generated automatically from the associated radiology reports. In this work the results from the CheXbert labeler (*Smit et al., 2020*) were used for the training labels.

The findings include *positive*, *negative*, *uncertain*, and *unmentioned (blank)* classes. In our experiments we used binary attribute classes, considering *unmentioned* and *negative* classes as class 0 and *positive* and *uncertain* as class 1. We considered all 13 provided attribute classes which include enlarged cardiomediastinum, cardiomegaly, lung opacity, lung lesion, edema, consolidation, pneumonia, atelectasis, pneumothorax, pleural effusion, other pleural, fracture, and support devices. The binary target classes used were the absence (0) or the presence (1) of one or more of the attribute findings.

Considering only samples with a frontal projection the total number of data samples is 191,229. Preprocessing includes the resizing of the images to a size of $64 \times 64$. The train and test split was adopted as defined in the dataset.

In contrast to the LIDC-IDRI dataset, the CheXpert dataset lacks segmentation masks, which is why the segmentation branch of the proposed method is disabled by excluding the respective term in the loss function.

## Method

### Model

The architecture comprises an encoder and several branches stemming from the output of the encoder, similar to previous work (*LaLonde, Torigian & Bagci, 2020*). The encoder is a capsule network consisting of convolutional layers that produce latent capsule vectors. These vectors serve as input to multiple heads that predict the target, attribute scores and prototypes, as well as a region of interest (ROI) mask.

The capsule vectors represent different features that have been extracted from the input image. Each feature is mapped to a single predefined attribute using supervised learning. This combination of the encapsulated feature representation and attribute mapping enables a truthful and human-understandable description of the decision process. Furthermore, the capsule information is used to generate attribute-specific prototypes that determine the attribute scores during inference and provide visual examples.

Referring to Fig. 1, the model can be divided into individual sections: The capsule values are calculated by the backbone, which is then followed by an attribute head, target head and segmentation head. The prototype branch extends the model by learning attribute-specific visual prototypes.

The *backbone* extracts features from the input image data and implements a capsule attention mechanism. Firstly the input data is processed by a convolutional layer containing 256 kernels of size 9 followed by a Rectified Linear Unit (ReLu) activation. A subsequent primary capsule layer applies another convolutional layer with 256 kernels of size 9 and segregates the resulting features in separate low level feature capsules. A final dense layer implements the dynamic routing algorithm (*Sabour, Frosst & Hinton, 2017*) in

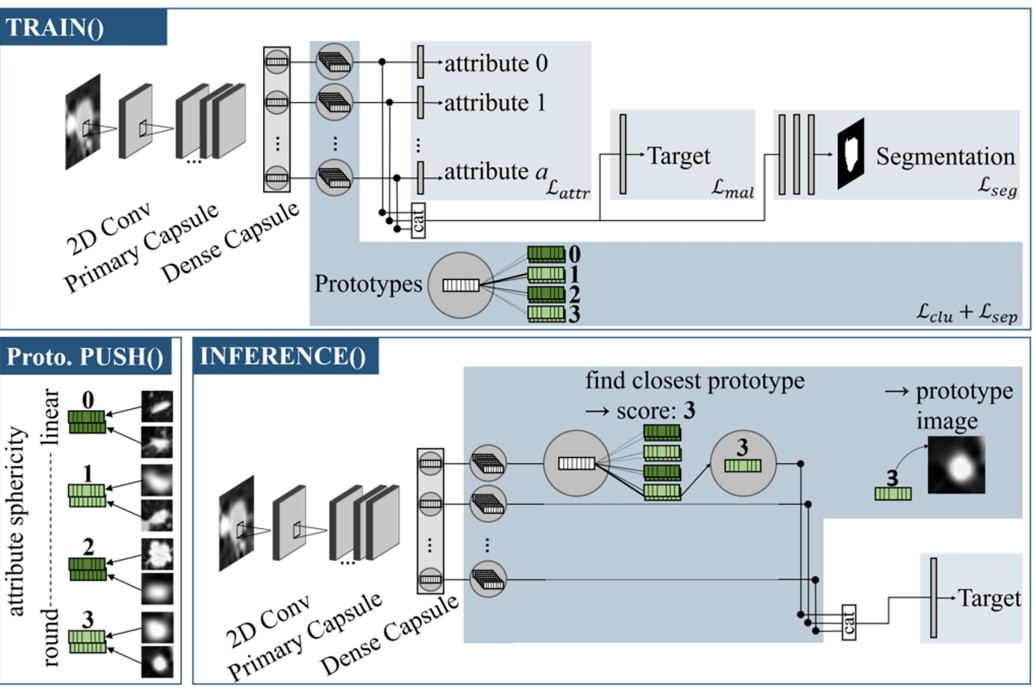

**Figure 1 Proto-Caps (here in 2D) is based on a capsule network.** TRAIN(): The capsule vectors are processed with four branches to optimize in regards of the attributes, target and segmentation mask (optional), and to learn prototype clusters. Prototype PUSH(): For each prototype vector of an attribute the sample from the training dataset with the smallest distance is saved. Here, prototypical images for the attribute *sphericity* are pushed, which have different degrees of roundness (from 0: linear to 3: circular). INFERENCE(): For each capsule vector, the closest prototype vector is found, leading to the attribute score and image. The closest prototype vectors also form the basis for the subsequent target prediction branch.

an iterative manner, resulting in 16-dimensional capsule vectors and forming the starting point for the following prediction branches.

The *target head* concatenates the capsule vectors and applies a fully connected layer. The output dimension is selected based on the label representation. For the LIDC-IDRI dataset, which has a distribution of five labels, the output dimension of the linear layer is five, followed by a softmax activation. For binary class labels, as in the CheXpert dataset, the output dimension is one with a sigmoid activation function.

The *segmentation head* decodes the capsule features to a mask of the ROI. The idea of learning this optional branch is an attention of the object to classify. The capsule vectors are concatenated and processed by two fully connected layers with 512 and 1,024 output features and ReLU activation. A final linear layer completes the upsampling and creates an output image of the same size as the original image and is followed by a sigmoid activation.

The *attribute head* implements the specialization of the capsules on individual, human-understandable attributes. Each capsule vector is processed by a separate linear layer and sigmoid activation to fit a respective attribute score.

The *prototype branch* contains multiple sample vectors for each capsule that eventually become feature vectors of prototypical samples during the training. For each attribute

value, a fixed number of prototypes are available to represent the visual variety of characteristics. The prototype vectors map real image samples and are visualized during inference.

The code of Proto-Caps is publicly available at https://github.com/XRad-Ulm/Proto-Caps. Algorithm 1 depicts the training and inference phases, described in the following.

### Training phase 1—feature extraction

At the beginning of the training, the model is optimized in solely respect of the training labels. The clustering of the prototypes is disabled at this stage. This step-wise approach allows for a prioritized focus on feature extraction.

In this first training phase, the network is optimized using the following combined loss function given the training labels.

$$\mathscr{L}_{P1} = \mathscr{L}_{tar} + \mathscr{L}_{attr} + \lambda_{seg} \cdot \mathscr{L}_{seg}. \tag{1}$$

The *target loss* $\mathscr{L}_{tar}$ refers to the target label and has been selected depending on the format of the label. In case of a distribution of the target annotations, as in the LIDC-IDRI dataset, the pointwise Kullback-Leibler divergence was used to reflect the inter-observer agreement and thus uncertainty (*LaLonde, Torigian & Bagci, 2020*). In the case of binary classification, as in the CheXpert dataset, a binary cross-entropy loss is applied.

$$\mathscr{L}_{tar} = Y_{tar} * \log \frac{Y_{tar}}{\hat{Y}_{tar}} \qquad (\text{LIDC} - \text{IDRI dataset})$$

$$\mathscr{L}_{tar} = BCE(Y_{tar}, \hat{Y}_{tar}) \qquad (\text{CheXpert dataset}). \tag{2}$$

The *attribute loss* $\mathscr{L}_{attr}$ is based on the error between the ground truth attribute score $Y_a$ and the network prediction $\hat{Y}_a$ for the $a$-th attribute:

$$\mathscr{L}_{attr} = b * \frac{1}{A} \sum_a^A \left\| Y_a - \hat{Y}_a \right\| \qquad (\text{LIDC} - \text{IDRI dataset})$$

$$\mathscr{L}_{attr} = b * \frac{1}{A} \sum_a^A BCE(Y_a, \hat{Y}_a) \qquad (\text{CheXpert dataset}). \tag{3}$$

If the attribute scores are continuous, as in the LIDC-IDRI dataset the mean square error is applied, else if the attribute scores are binary, as in the CheXpert dataset, a binary cross-entropy loss is applied. The random binary mask $b$ can be used to control from which training sample the attribute annotations are used and from which not, allowing semi-supervised attribute learning. This version was investigated in case attribute labels are only available for part of the training dataset (see "Sparse Data Study").

The *segmentation loss* $\mathcal{L}_{seg}$ is used when the dataset includes segmentation masks of the region of interest. By incorporating this loss term the segmentation branch of the model is activated with the mean square error between the reconstructed and the ground truth mask.

$$\mathscr{L}_{seg} = \left\| Y_{mask} - \hat{Y}_{mask} \right\|. \tag{4}$$

**Algorithm 1**  Algorithm of proto-caps.

$\vec{c}^a$: capsule vectors $a = 1, ..., A$

$\vec{C}$: concat. of capsule vectors $\vec{C} = (\vec{c}^1, \vec{c}^2, ..., \vec{c}^A)$

$m_{branch}(input)$: process input with layer(s) of *branch*

TRAIN()

    $\hat{Y}_a \leftarrow m_{attribute^a}(\vec{c}^a)$

    $\hat{Y}_{target} \leftarrow m_{target}(\vec{C})$

    $\hat{Y}_{mask} \leftarrow m_{segmentation}(\vec{C})$

    IF epoch $<$ *warmup*:

        BACKPROPAGATE($\mathscr{L}_{P1}$)

    ELSE:

        BACKPROPAGATE($\mathscr{L}_{P2}$)

        every $(pushstep)^{th}$ epoch:

            PUSH()

$\vec{p}^{a,p}$: prototype vectors $p = 1, ..., nP$

PUSH()

    Iterate over $N$ training samples:

        Calculate distance $||\vec{c}^a - \vec{p}^{a,p}||$

    For each prototype vector, save closest training sample

INFERENCE()

    $\vec{C}_{proto} \leftarrow (\vec{p}^{1,p_1}, \vec{p}^{2,p_2}, ... \vec{p}^{A,p_A})$, where $p_i$ is index of

                            closest $\vec{p}^{a,p}$ for each $a$

    $\hat{Y}_a \leftarrow p^{a,p_a}$ (attribute score of closest prototype)

    $\hat{Y}_{target} \leftarrow m_{target}(\vec{C}_{proto})$

The hyper-parameter $\lambda_{seg} = 0.512$ was chosen according to *LaLonde, Torigian & Bagci (2020)*.

### Training phase 2—training prototypes

After the warm-up phase, the randomly initialized prototype vectors are being unfrozen and trained to represent cluster centers for the differen attribute manifestations. Motivated by existing work about prototype learning (*Chen et al., 2019*), the combined loss function is being extended by two terms, defined as a cluster loss $\mathscr{L}_{clu}$ and a separation loss $\mathscr{L}_{sep}$. The cluster loss reduces the Euclidean distance between the capsule vector $\vec{c}^a$ and the nearest prototype $\vec{p}^{a,p}$ of the correct attribute score in $P_{a_s}$.

$$\mathscr{L}_{clu} = \frac{1}{A} \sum_{a}^{A} \min_{\vec{p}^{a,p} \in P_{a_s}} \left\| \vec{c}^a - \vec{p}^{a,p} \right\|_2. \tag{5}$$

On the other hand the separation loss increases the distance between the capsule vector and prototypes, that are dedicated to other attribute scores, limited by a maximum distance:

$$\mathscr{L}_{sep} = \frac{1}{A} \sum_{a}^{A} \min_{\vec{p}^{\,a,p} \notin P_{a_s}} \max(0, \mathrm{dist}_{max} - \left\| \vec{c}^{\,a} - \vec{p}^{\,a,p} \right\|_2). \tag{6}$$

The overall loss function for the second training phase is the following weighted sum, where the separation factor $\lambda_{sep} = 0.1$ was chosen empirically:

$$\mathscr{L}_{P2} = \mathscr{L}_{tar} + \mathscr{L}_{attr} + \lambda_{seg} \cdot \mathscr{L}_{seg} + \mathscr{L}_{clu} + \lambda_{sep} \cdot \mathscr{L}_{sep}. \tag{7}$$

For a visual representation of the optimized prototype vectors with real sample images, training samples are pushed onto the prototype vectors. Two approaches of this push operation were examined. The first approach includes to find the sample with the smallest distance for each prototype vector and save it with the original image and the ground truth information. The second approach, which is similar to existing work about prototype learning (*Chen et al., 2019*) additionally involves replacing the prototype vector with the vector generated by this real sample.

The repetition rate of this operation is set by the hyper-parameter *pushstep*.

### Inference phase—predict from prototypes

During inference, the prediction of the attribute and target scores is based on the sample prototypes. Firstly, the prototype vectors that are closest to the capsule vectors are determined. The attribute prediction is set to the attribute label of the corresponding closest prototype. The trained attribute layers are ignored in the inference phase. For the target prediction, the prototype vectors are concatenated and processed by the prediction layer.

## EXPERIMENTS AND RESULTS

### Model parameters

The model layers were optimized using an Adam optimizer with a learning rate of 0.02. The iterative dynamic routing algorithm was implemented in three iterations. Training phase 2, where the prototypes are learned, starts after the warm-up phase. The warm-up phase consists of 100 epochs for the LIDC-IDRI dataset and of one epoch for the CheXpert dataset. The push operations repetition rate was set to *pushstep* = 10 for the LIDC-IDRI dataset and to *pushstep* = 1 for the CheXpert dataset. In preliminary studies, we initiated the warm-up phase once the loss $\mathscr{L}_{P1}$ had converged. We also selected the hyperparameter *pushstep* by balancing training time with the stabilization of individual loss components.

With a maximum of 1,000 epochs, but stopping early if there was no improvement in the mean of target and attribute accuracy within 10 push steps, the experiments lasted an average of 3 h on a GeForce RTX 3090 graphics card for the LIDC-IDRI dataset in the 2D setting. The 3D LIDC-IDRI experiments took an average of 14 h due to the larger memory

requirements of the prototypes. The CheXpert experiments took longer per epoch due to the larger size of the input and dataset, and training was stopped early after 48 h.

The number of capsules used in the architecture was chosen in relation to the number of attributes, *i.e.*, eight capsules for the LIDC-IDRI dataset and 13 capsules for the CheXpert dataset.

## Explainability

Proto-Caps is a xAI method that is explainable by design. Unlike *post-hoc* explainable AI, which requires a second algorithm to provide explanations, this intrinsically explainable method offers interpretability within the inference process.

The network uses the closest prototypical vectors to calculate attribute and target predictions. These vectors are linked to data samples from the training dataset. This allows for the visualization of prototypes with real images to validate the prediction results. Figure 2 illustrates the use of generated attribute prototypes to explain the model prediction.

Figure 2 presents three example cases (A, B, and C), each shown in a separate row. Case A demonstrates a correct target prediction (malignancy), while Cases B and C highlight incorrect predictions. To explain the model's decision, prototypes for three attributes (margin, lobulation, and spiculation) are shown. Since the target prediction is based on these attribute prototypes, the reasoning behind the model's decision can be expressed as follows: "The model predicted malignancy 4 because the detected features in the input image closely resemble this prototype for the attribute *margin*, and this prototype for the attribute *lobulation*, *etc.*". This interpretability allows for result validation: if discrepancies are observed between the input image and the retrieved prototypes, it may indicate potential model errors. Such insights can help identify cases where the prediction should be questioned.

### *Faithfulness*

The following experiments investigate the extent to which the explanations reflect the model's decision.

In the first experiment we analyzed the *correlation between attributes and target correctness* using a *post-hoc* logistic regression study: Based on the correctness of the attribute prediction, the correctness of the target prediction was calculated. The result on the LIDC-IDRI dataset shows a strong relationship between both with a prediction accuracy of 94.1% /1.1.

In the second experiment we analyzed the *joint feature importance* for the combined task in order to gain insights about the information flow within the network. The explainability of Proto-Caps relies on the capsule vectors as a foundation for decision-making. These are used to derive the attribute scores and also to classify the target. To ensure the trustworthiness of the explanations, we used the LIDC-IDRI dataset to analyze whether the information was shared across the multidimensionality of the capsule vectors for this combined task.

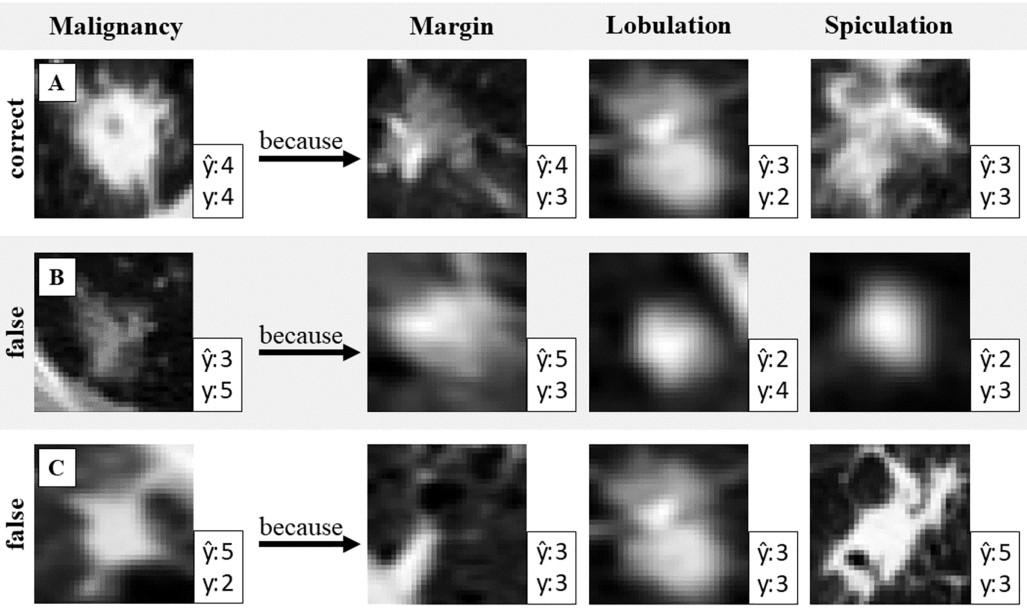

**Figure 2 Validation of the malignancy prediction.** One correct and two wrongly predicted examples with attribute prototypes. Prediction $\hat{y}$ and ground truth label y of malignancy and attribute respectively. Identifying false attribute predictions can help to identify misclassification in malignancy.

The joint information is derived from the weights $w$ of the linear layer of each attribute and the target prediction. The layer input values are used to standardize the weights $w'$. To achieve this, the capsule vectors $z$ for the test dataset are first standardized:

$$z' = \frac{z - \tilde{z}}{\sigma_z}. \tag{8}$$

The standardized weights are calculated using the linear function:

$$y = z * w = z' * w'. \tag{9}$$

The correlation functions are calculated from each attribute layer $w'^i_{attr}$ and the respective weight values of the target layer $w'^i_{tar}$:

$$\text{corr}^i_P = \text{Pearson}(|w'^i_{tar}|, |w'^i_{attr}|) \tag{10}$$

$$\text{corr}^i_S = \text{Spearman}(|w'^i_{tar}|, |w'^i_{attr}|). \tag{11}$$

The mean correlation over the eight attribute capsules and five target classes for the LIDC-IDRI dataset is with the Pearson calculation 0.62 and with Spearman 0.54. These values show a moderate to strong correlation.

The experiments show that the model's decision making process is largely driven by the attribute explanation.

### User-centered evaluation

In addition to the technical investigation of the explanations, we conducted a user-centered evaluation of the attribute explanations provided by Proto-Caps

(*Gallée et al., 2024*). Six radiologists diagnosed lung nodules from the LIDC-IDRI dataset in test scenarios while having access to the model output. For each case, three variants of explanation were presented: (A) the malignancy prediction alone, (B) the prediction with attribute scores or (C) the prediction with both attribute scores and attribute prototypes.

The evaluation incorporated objective measures—such as the diagnostic accuracy of the radiologists across the different model support variants—as well as subjective assessments of trust and perceived helpfulness.

Figure 3 illustrates that the explanations provided by Proto-Caps are persuasive: when the model prediction is correct, they lead to higher diagnostic accuracy, whereas in cases of incorrect predictions, the explanations tend to increase the likelihood of an incorrect diagnosis. Radiologists reported that the explanations aligned with their diagnostic criteria and served as useful guidance during diagnosis.

To mitigate these negative effects, we suggest using Proto-Caps selectively—primarily during development and testing—and, in practice, as a warning system that activates and intervenes only when the model suspects a false negative diagnosis by the radiologist. For a thorough discussion of the experimental design, survey methodology, and analysis of user feedback, readers are referred to *Gallée et al. (2024)*.

## Performance

### Evaluation metric

For the LIDC-IDRI dataset Proto-Caps is evaluated using the Within-1-Accuracy metric. A prediction is considered correct if it falls within a tolerance of 1 score of the ground truth label. As in previous work on the LIDC-IDRI dataset (*LaLonde, Torigian & Bagci, 2020*), this metric is used for the ordinal labels of malignancy (ascending from 1-*highly unlikely* to 5-*highly likely*).

For the CheXpert dataset, where possible, area under the curve (AUC) values are calculated as an evaluation measure to compare the results of Proto-Caps with other studies. However, the calculation by including prediction probabilities in the Proto-Caps evaluation is only possible for the target classification, not for the attribute classification. This is due to the prototype-based inference for the attributes. The predicted attribute classification is determined by the ground truth scores of the closest prototypes.

### Comparison with SOTA

We compare our proposed method with literature values of existing methods. For the LIDC-IDRI dataset these include methods that predict either no attributes (non-explainable) or some attributes (explainable). Table 1 displays the accuracy of the 2D models, while Table 2 presents the accuracy of the 3D models.

The results show that Proto-Caps outperforms existing explainable approaches in predicting both nodule malignancy and visual attributes. X-Caps (*LaLonde, Torigian & Bagci, 2020*) in Table 1 is also a capsule-based architecture similar to Proto-Caps. It first learns attributes and then predicts malignancy based on them. However, it does not learn attribute prototypes.

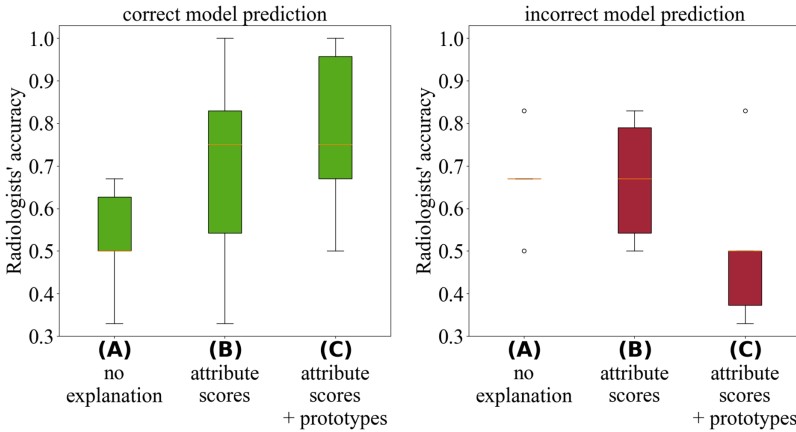

**Figure 3** **Analysis by** *Gallée et al. (2024)* **of the radiologists' performance during the test cases.** The green boxplots (left) depict diagnostic accuracy when the model prediction was correct, while the red boxplots (right) show accuracy when the model prediction was incorrect.

**Table 1** **2D Comparison with literature values of other works, attribute scores are reported if available.** Mean $\mu$ and standard deviation $\sigma$ calculated from 5-fold experiments. Scores reported in Within-1-Accuracy. The best result is in bold.

| | | Attribute Prediction Accuracy in % | | | | Malig-nancy | | | | |
|---|---|---|---|---|---|---|---|---|---|---|
| | | Sub | IS | Cal | Sph | Mar | Lob | Spic | Tex | |
| **Non-explainable** | | | | | | | | | | |
| 3D-CNN+MTL (*Hussein et al., 2017a*) | | – | – | – | – | – | – | – | – | 90.0 |
| TumorNet (*Hussein et al., 2017b*) | | – | – | – | – | – | – | – | – | 92.3 |
| CapsNet (*LaLonde, Torigian & Bagci, 2020*) | | – | – | – | – | – | – | – | – | 77.0 |
| **Explainable** | | | | | | | | | | |
| X-Caps (*LaLonde, Torigian & Bagci, 2020*) | | 90.4 | – | – | 85.4 | 84.1 | 70.7 | 75.2 | **93.1** | 86.4 |
| Proto-Caps (proposed) | $\mu$ | **91.4** | **99.4** | **96.9** | **92.5** | **86.7** | **88.3** | **89.0** | **93.1** | **93.1** |
| | $\sigma$ | 2.3 | 0.7 | 1.2 | 4.6 | 4.3 | 2.8 | 2.4 | 1.0 | 0.9 |

**Table 2** **3D Comparison with literature values of other works, attribute scores are reported if available.** Mean $\mu$ and standard deviation $\sigma$ calculated from 5-fold experiments. Scores reported as Within-1-Accuracy, except for *Mehta et al. (2021), Afshar et al. (2020), Zhu et al. (2018), Shen et al. (2019)* reporting binary AUC (marked with asterix*), and binary ACC (marked with [+]) respectively.

| | | Attribute prediction accuracy in % | | | | Malig-nancy | | | | |
|---|---|---|---|---|---|---|---|---|---|---|
| | | Sub | IS | Cal | Sph | Mar | Lob | Spic | Tex | |
| **Non-explainable** | | | | | | | | | | |
| CNN+Rand. Forest (*Mehta et al., 2021*)* | | – | – | – | – | – | – | – | – | 80.7 |
| Deeplung (*Zhu et al., 2018*)[+] | | – | – | – | – | – | – | – | – | 90.4 |
| 3D-MCN (*Afshar et al., 2020*)* | | – | – | – | – | – | – | – | – | 96.4 |
| **Explainable** | | | | | | | | | | |
| HSCNN (*Shen et al., 2019*)[+] | | 71.9 | – | 90.8 | 55.2 | 72.5 | – | – | 83.4 | 84.2 |
| 3D Proto-Caps (proposed) | $\mu$ | 92.5 | 99.8 | 98.2 | 96.7 | 93.9 | 93.1 | 93.2 | 95.3 | 94.3 |
| | $\sigma$ | 3.7 | 0.2 | 1.2 | 2.1 | 2.2 | 1.7 | 2.5 | 1.3 | 1.7 |

Compared to methods that do not implement attribute prediction for improved explainability (3D-CNN+MTL, TumorNet, CapsNet), our method performs better in the 2D case. In the 3D case, as shown in Table 2, it is important to note that the data preprocessing and evaluation metrics differ among the compared methods. Our proposed method achieves a higher malignancy prediction accuracy in the 3D setting (2D: 93.1 *vs.* 3D: 94.3) as well as a better mean prediction accuracy for the eight attributes (2D: Ø = 92.2 *vs.* 3D: Ø = 95.3). These results demonstrate the methodological feasibility of Proto-Caps on 3D data.

Proto-Caps also excels in the second validation dataset, CheXpert, in predicting fractures, pneumothorax, pneumonia and lung lesions, but underperforms in other attributes, see Table 3.

## Hyperparameter evaluation

We analyzed various methodologically crucial parameters of the model to show which parameters the model is sensitive to and to provide a starting point for future research with the model. These parameters include the size of the prototype set per attribute, the dimension of the capsule vectors, and the number of capsule vectors. Furthermore, we conducted an investigation into variants of the prototype layer's push operation. The experiments were conducted on the LIDC-IDRI dataset.

### *Number of prototypes*

In the following we present the results of the experiments where we used a different number or prototypes that are initialized per attribute class. Besides the original setting with 16 prototypes per attribute class, we tested four, 32 and 64 prototypes per attribute class. The results are summarized in Fig. 4 and in the following:

Accuracy: In terms of the target accuracy 93.4/1.2 (4) 93.1/0.9 (16), 92.9/1.4 (32), 93.4/1.24 (64), and of the mean attribute accuracy 92.9/1.7 (4) 92.2/2.4 (16), 92.3/2.5 (32), 91.9/3.6 (64), there is no advantage in using more prototypes.

Training time: When changing the number of prototypes, the survey of the training time needs to be considered. No change in duration is detected during the weight adjustment phase (18 s). However, one push operation of the prototypes takes significantly longer the more prototypes are learned: 17 s (four), 37 s (16), 51 s (32), and 69 s (64).

Number of prototypes used in inference: This experiment analyzed the total number of attribute prototypes actually used in the test dataset: 107/160 (4) 311/640 (16), 418/1,280 (32), and 569/2,460 (64).

Having a diverse set of prototypical samples used on the testing dataset increases the quality of the local explainability meaning that the prototypes are adapted better on the different manifestations within one attribute class. A small set of prototypes provides a global explanation of the entire network, while a large number of prototypes offers a more local explanation for each sample.

High target and attributes accuracies were achieved when using four, 16, and 32 prototypes per attribute class. As the average number of prototypes (16 per attribute class,

**Table 3 Comparison of the experiment on the CheXpert dataset with literature values of another work.** The evaluation metric used by Proto-Caps is the ACC for the attributes (marked with an asterisk), while it uses the area under the curve (AUC) as the performance metric for the target and for the comparison work.

|  | CheXGCN (*Chen et al., 2020*) | Proto-Caps (proposed) |
|---|---|---|
| Attributes |  |  |
| Enlarged cardiomed. | 69.7 | 48.0* |
| Cardiomegaly | 87.7 | 74.8* |
| Lung opacity | 82.2 | 69.8* |
| Lung lesion | 76.8 | 99.5* |
| Edema | 88.6 | 80.2* |
| Consolidation | 78.4 | 84.2* |
| Pneumonia | 81.0 | 96.0* |
| Atelectasis | 73.6 | 70.8* |
| Pneumothorax | 91.7 | 96.5* |
| PleuralEffusion | 90.7 | 69.3* |
| Pleural other | 83.5 | 99.5* |
| Fracture | 83.3 | 100.0* |
| Support device | 89.9 | 59.9* |
| Target (normal/abnormal) | 87.9 | 87.4 |

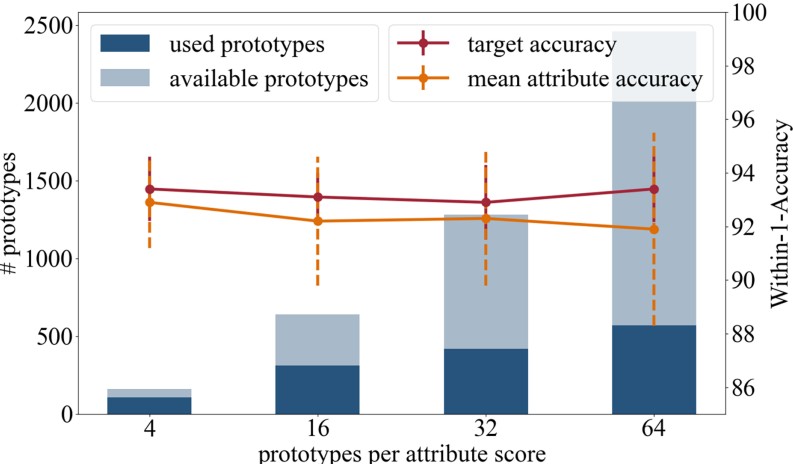

**Figure 4 Analysis in regards of the number of prototypes available per attribute score.**

*i.e.,* 80 per attribute) reflects the variance of the appearance well, we recommend the use of 16 prototypes per attribute class.

### Capsule vector dimension

This section of experiments explores the effect of the dimension of the attribute capsules. The motivation for reducing the dimension is to reduce the number of learnable parameters, which raises the prospect of a more generalized representation of the

attributes. In addition to the original capsule setting of 16 dimensional vectors, dimensions 2, 4, 8 and 32 were tested.

Accuracy: In terms of the target accuracy 93.5/1.3 (2), 93.0/1.6 (4), 93.3/0.9 (8), 93.1/0.9 (16), 92.6/0.9 (32), and of the mean attribute accuracy 93.1/2.5 (2), 93.1/2.0 (4), 92.3/2.2 (8), 92.2/2.4 (16), 91.0/3.4 (32), the prediction accuracies are comparable.

Number of prototypes used in inference: Of the 640 available prototypes during inference 181 (2), 265 (4), 330 (8), 311 (16), and 265 (32) were used.

Based on high prediction accuracy and the number of prototypes used, see Fig. 5, we recommend a capsule dimension of 8 or 16, as similar results were achieved with both.

### Number of capsules

In previous work (*LaLonde, Torigian & Bagci, 2020*), the number of capsules was set to the number of attributes. This model architecture supports the idea that the target prediction is based on the defined attribute capsules. Since one could allow the prediction to be based on more than just the predefined attributes, we investigated how the model changes when the model has more capsules. The model architecture is changed by adding additional capsules that are used for the target prediction branch but are not followed by attribute prediction layers. For this experiment, we tested the variants with 4, 8, and 16 extra capsules.

Accuracy: The target's accuracy was 78.9/10.9 (+4), 90.4/1.1 (+8), and 89.0/3.8 (+16) compared to the original accuracy of 93.1/0.9 (+0) when no additional capsules were used besides the supervised attribute capsules. The mean attribute accuracy was 91.2/2.9 (+4), 92.2/2.4 (+8), and 92.0/2.3 (+16) compared to 92.2/2.4 with the original setting.

The results indicate a decrease in target accuracy and a reduced level of robustness with the inclusion of additional capsules.

### Push operation

In the following experiment on the LIDC-IDRI dataset, we compare two versions of the push operation. This operation occurs during training phase 2, when the prototype branch is being optimized. For each prototype vector, the push operation stores a real sample image to which its capsule vector is closest.

In the first variant, which was used in the experiments above, the push operation does not involve any additional steps.

In the second variant, which is based on the ProtoPNet prototype learning strategy by *Chen et al. (2019)*, the capsule vectors of the most similar image samples replace the prototype vectors in the push operation. This additional step changes the learning progress of the prototype vectors, as they are not only adapted by the loss function, but are also replaced during the push operations.

The push variants were tested with the original capsule dimension of 16 and with 8.

It was observed that the training accuracy after the first push operation is higher with the replacing-push method. By replacing the prototype vectors with real samples, they are adapted faster initially, in contrast to the slow adaptation caused by only the change from backpropagation.

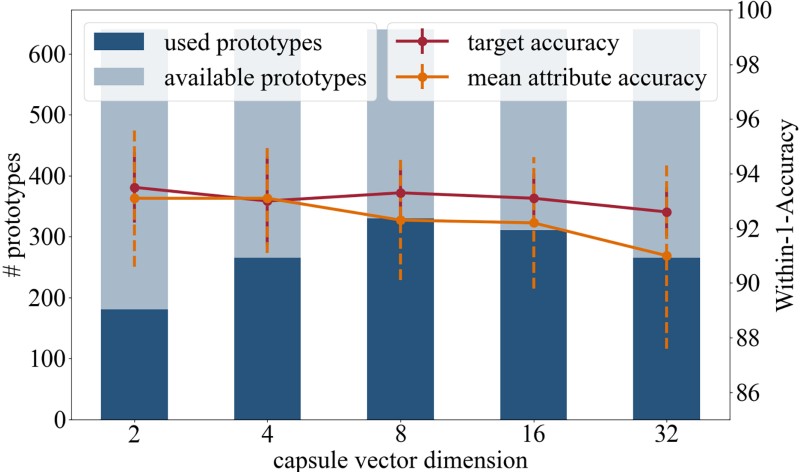

**Figure 5** **Analysis in regards of the dimension of the capsule vectors.**

With the replacing-push method, the final accuracy for target prediction is 93.3/1.2 (16), and for mean attribute prediction it is 92.5/2.4 (16) for the capsule dimension 16. For the capsule dimension of 8 the target prediction accuracy is 90.5/2.1 (8) and the mean attribute prediction accuracy is 92.9/1.7 (8).

When comparing accuracies, there are no differences between the proposed push operation and the replacement-push variant, except for a lower target prediction accuracy when using a capsule vector dimension of 8. We recommend using the proposed push variant, where the prototype vectors are fitted through backpropagation with the prototype loss function only.

## Sparse data study

Our method creates prototypes that represent human understandable features. These features are predefined and integrated into the model through training with attribute labels. The supervised approach to attribute learning thus places particular demands on the training dataset. In the case of a medical application, this implies a higher and therefore also more costly annotation effort due to the necessity of medical expertise.

To increase the applicability of the proposed method to other datasets and reduce the required annotation effort, we conducted tests using fewer attribute labels. For this purpose, we trained the model using only the attribute annotations of a small fraction (10% = 1,907 samples and 1% = 190 samples) of the LIDC-IDRI dataset. We used only the target and segmentation labels for the remaining data samples. Table 4 shows that good results can be achieved even with reduced availability of attribute annotations.

Additionally, we examined the behavior of target prediction accuracy when none of the attribute labels are used, resulting in capsule vectors that are not mapped on predetermined attributes. A similar test accuracy demonstrates that the additional restriction of mapping defined features to capsule vectors has no negative impact on model performance.

**Table 4 Results of the sparse data study. Attribute annotations were used from a fraction of the training dataset.** Mean $\mu$ and standard deviation $\sigma$ calculated from 5-fold experiments. Scores reported as Within-1-Accuracy.

| | | Attribute prediction accuracy in % | | | | Malignancy | | | | |
|---|---|---|---|---|---|---|---|---|---|---|
| | | Sub | IS | Cal | Sph | Mar | Lob | Spic | Tex | |
| 100% attribute labels | $\mu$ | 91.4 | 99.4 | 96.9 | 92.5 | 86.7 | 88.3 | 89.0 | 93.1 | 93.1 |
| | $\sigma$ | 2.3 | 0.7 | 1.2 | 4.6 | 4.3 | 2.8 | 2.4 | 1.0 | 0.9 |
| 10% attribute labels | $\mu$ | 94.1 | 99.8 | 96.5 | 96.9 | 93.0 | 92.1 | 90.7 | 93.9 | 92.7 |
| | $\sigma$ | 1.5 | 0.3 | 0.9 | 1.4 | 1.2 | 2.6 | 1.6 | 2.4 | 1.5 |
| 1% attribute labels | $\mu$ | 95.6 | 99.8 | 96.1 | 97.0 | 91.1 | 90.8 | 88.6 | 93.3 | 92.3 |
| | $\sigma$ | 1.0 | 0.3 | 1.9 | 1.3 | 1.0 | 2.8 | 1.3 | 1.9 | 1.6 |
| 0% attribute labels | $\mu$ | – | – | – | – | – | – | – | – | 92.4 |
| | $\sigma$ | – | – | – | – | – | – | – | – | 1.0 |

It was demonstrated that the proposed method can be successfully employed in the absence of attribute labels. Although the prototype explanations are not defined more precisely, they still resemble the recognized features.

## DISCUSSION

We propose a deep neural network for image classification with a specialized architecture that enhances human interpretability by enabling users to understand its information processing. The network uses a hierarchical data process to establish truthful causality between human-defined decision criteria and the final classification.

Validation with visual prototypes, which are specific in terms of distinctive, previously defined features, mimics the decision-making process of humans, offering a logical reasoning framework that increases trust in the model and helps identify misclassifications. The strong correlation between misclassified attributes and target confirms the model's coherence.

Despite the extended explainability of our method, it does not compromise accuracy, as demonstrated by comparisons with other studies. Experiments conducted on the LIDC-IDRI dataset demonstrated that Proto-Caps outperforms existing explainable methods and achieves similar results to non-explainable approaches. This establishes Proto-Caps as a new state-of-the-art, incorporating all relevant attributes that human experts deem important.

It is important to note that the datasets used come with inherent limitations. *Zhang et al. (2022)* note that subjective radiologist assessments in the LIDC-IDRI database can introduce label errors and supervision bias. Nonetheless, the dataset remains a large-scale and widely used benchmark for lung cancer prediction. Additionally, *Baltatzis et al. (2021)* highlight that data selection affects class distribution, an issue we mitigate by using the same preprocessing as prior work (*LaLonde, Torigian & Bagci, 2020*; *Smit et al., 2020*). For CheXpert, studies by *Glocker et al. (2023)* and *Seyyed-Kalantari et al. (2020)* reveal biases in model performance across race and sex.

Additionally, an implementation of Proto-Caps using a 3D convolutional network as its backbone demonstrates competitive results, further validating the model's effectiveness across different settings.

Given the limited availability and high cost of creating medical datasets with both visual attributes and target classifications, this work explores alternatives to a fully labeled dataset. Our experiments show that even a small fraction of attribute annotations, as little as 190 samples in the training dataset, is adequate to generate attribute-specific prototypes with a high degree of accuracy. This makes the method applicable to other datasets with minimal additional annotation effort. *Zhang et al. (2022)* note that subjective radiologist assessments in the LIDC-IDRI database can introduce label errors and supervision bias. However, the dataset is large-scale and widely used as a benchmark for lung cancer prediction. *Baltatzis et al. (2021)* highlight that data selection affects class distribution, but we mitigate this issue by using the same preprocessing as prior work (*LaLonde, Torigian & Bagci, 2020*; *Smit et al., 2020*). For CheXpert, studies by *Glocker et al. (2023)* and *Seyyed-Kalantari et al. (2020)* reveal biases in model performance across race and sex.

Various components of the model were investigated, revealing a fair joint feature importance between the target and attribute predictions. Furthermore, several model parameters were optimized using the LIDC-IDRI dataset, with the primary objective of achieving local explainability by utilizing a large number of prototypes. The selection of various dimensions and quantities helped identify the optimal model configuration.

Validating an AI method that claims to be explainable is challenging, as it requires not only technical model examination but also an analysis of the explanations' truthfulness, helpfulness, and predictive accuracy. In addition to model evaluation, it is critical to involve potential end-users when assessing explainable AI methods. This is especially true in sensitive areas such as medicine, where complex tasks demand tailored solutions.

User studies are an essential part of this process, ideally conducted early in AI development to ensure the AI aligns with user needs. A user study conducted with radiologists using Proto-Caps has demonstrated that explanations have an impact on their decision-making. The model's confidence increases with the number of arguments presented in favor of its decision. While there was a tendency for human performance to improve with more explanations when the model was correct, there was also a negative tendency for users to be led astray by more explanations and make the wrong decision when the model was incorrect. This finding highlights the impact of model explanations on users and raises important questions about human-AI interaction. It also suggests that AI can potentially enhance human performance, but it requires careful consideration of the design of explanations.

While the results of our study are promising, real-world deployment of this approach faces additional challenges that need further research. For instance, domain-shift adaptation will be critical to ensure robustness when the model encounters unseen data from different settings. Moreover, testing the model with other medical imaging modalities, such as magnetic resonance imaging (MRI) and ultrasound, would be essential to evaluate its generalizability. Finally, exploring its application to a broader range of pathologies will be important to assess its robustness across diverse medical contexts.

These avenues for future work will be crucial to making the method more practical and effective in real-world clinical settings.

## CONCLUSION

As research into deep neural networks in medical applications progresses, it is crucial to consider the safety of AI models. To ensure safe usage, interpretability of the models is necessary. If model interpretability is considered during development, decision support systems can be created that are intrinsically explainable and allow for human-understandable validation. This work shows an example of the inclusion of additional information and its processing, which makes the model more interpretable. Prototypes are used to identify similarities to the inference sample based on predefined attributes.

The explanation follows the user's intuition using attributes that are understandable and by providing a visual comparison. By imitating the decision-making process of humans, the user can discuss the model's prediction and evaluate how much they trust it.

## ACKNOWLEDGEMENTS

The authors used ChatGPT to check their grammar.

### Funding

This research was supported by the University of Ulm (Baustein, L.SBN.0214), and the German Federal Ministry of Education and Research (BMBF) within RACOON COMBINE "NUM 2.0" (FKZ: 01KX2121), and PC3-AIDA-Advanced Imaging Utilization by Digital Data. The funders had no role in study design, data collection and analysis, decision to publish, or preparation of the manuscript.

### Grant Disclosures

The following grant information was disclosed by the authors:
University of Ulm (Baustein, L.SBN.0214).
German Federal Ministry of Education and Research (BMBF).
RACOON COMBINE "NUM 2.0": FKZ: 01KX2121.
PC3-AIDA-Advanced Imaging Utilization by Digital Data.

### Competing Interests

The authors declare that they have no competing interests.

### Author Contributions

- Luisa Gallée conceived and designed the experiments, performed the experiments, analyzed the data, performed the computation work, prepared figures and/or tables, authored or reviewed drafts of the article, and approved the final draft.
- Catharina Silvia Lisson conceived and designed the experiments, authored or reviewed drafts of the article, and approved the final draft.

- Timo Ropinski conceived and designed the experiments, authored or reviewed drafts of the article, and approved the final draft.
- Meinrad Beer conceived and designed the experiments, authored or reviewed drafts of the article, and approved the final draft.
- Michael Götz conceived and designed the experiments, analyzed the data, authored or reviewed drafts of the article, and approved the final draft.

### Data Availability

The code is available at GitHub: https://github.com/XRad-Ulm/Proto-Caps.

The LIDC-IDRI dataset is available at: https://www.cancerimagingarchive.net/collection/lidc-idri.

A Large Chest X-Ray Dataset And Competition is available at: https://stanfordmlgroup.github.io/competitions/chexpert.

CheXbert is available at GitHub: https://github.com/stanfordmlgroup/CheXbert.

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
