# Peer review of "Proto-Caps: interpretable medical image classification using prototype learning and privileged information"

_PeerJ Computer Science, doi:10.7717/peerj-cs.2908_

## Round 0.1 · original submission · Major Revisions

The authors must improve the paper as per the suggestions of the reviewers, and also add more critical discussions.

Reviewer 1 ·

Basic reporting

The manuscript introduces Proto-Caps, a deep learning-based framework designed to provide interpretable classifications for medical images. The method integrates prototype learning with privileged information, leveraging capsule networks to ensure human-understandable explanations. Experimental results on LIDC-IDRI and CheXpert datasets demonstrate its effectiveness compared to other explainable AI approaches while maintaining competitive classification performance. While the study highlights innovative aspects, several areas require improvement to strengthen its contribution and practical significance.

• The author should include a concise description of how Proto-Caps improves interpretability over existing methods in the introduction section.
• It is suggested to address the limitations of datasets such as lack of diversity and possible biases. Discuss how these limitations might impact the model's generalizability.
• To further strengthen the introduction, I recommend incorporating and discussing the following studies: 1: A knowledge-based image enhancement and denoising approach 2: Optimal feature extraction and ulcer classification from WCE image data using deep learning 3: Contrast Enhancement of Low-Contrast Medical Images Using Modified Contrast Limited Adaptive Histogram Equalization.
• Provide more examples of prototypes and their corresponding input images in various scenarios to better illustrate their interpretability.
• Discuss how the prototypes handle ambiguous cases or noise in the data.
• Include comparisons with state-of-the-art methods in explainable AI, specifically for capsule-based architectures.
• The manuscript mentions validation of explanations through radiologists but lacks a comprehensive evaluation of the explanation quality. Include metrics or user study results that assess the interpretability and trustworthiness of Proto-Caps explanations.

Experimental design

No comment

Validity of the findings

No comments

Additional comments

The manuscript introduces Proto-Caps, a deep learning-based framework designed to provide interpretable classifications for medical images. The method integrates prototype learning with privileged information, leveraging capsule networks to ensure human-understandable explanations. Experimental results on LIDC-IDRI and CheXpert datasets demonstrate its effectiveness compared to other explainable AI approaches while maintaining competitive classification performance. While the study highlights innovative aspects, several areas require improvement to strengthen its contribution and practical significance.

• The author should include a concise description of how Proto-Caps improves interpretability over existing methods in the introduction section.
• It is suggested to address the limitations of datasets such as lack of diversity and possible biases. Discuss how these limitations might impact the model's generalizability.
• To further strengthen the introduction, I recommend incorporating and discussing the following studies: 1: A knowledge-based image enhancement and denoising approach 2: Optimal feature extraction and ulcer classification from WCE image data using deep learning 3: Contrast Enhancement of Low-Contrast Medical Images Using Modified Contrast Limited Adaptive Histogram Equalization.
• Provide more examples of prototypes and their corresponding input images in various scenarios to better illustrate their interpretability.
• Discuss how the prototypes handle ambiguous cases or noise in the data.
• Include comparisons with state-of-the-art methods in explainable AI, specifically for capsule-based architectures.
• The manuscript mentions validation of explanations through radiologists but lacks a comprehensive evaluation of the explanation quality. Include metrics or user study results that assess the interpretability and trustworthiness of Proto-Caps explanations.

·

Basic reporting

- Clarity and Language: The paper is written in clear, professional English. However, some sentences, particularly in the introduction and related work sections, are overly verbose and vacuous, which could hinder readability.

- Background and References: While the manuscript references relevant prior works, the citation density in some sections (e.g., "Privileged Information in Capsule Network") appears excessive without adding clarity (i.e., overly verbose and vacuous). Additionally, there is little discussion on the limitations of competing methods, which would contextualize the authors’ contributions more effectively.

- Figures and Tables: The figures and tables are well-presented and relevant. However, Figure 1 ("Proto-Caps architecture") is visually dense and requires additional explanation for clarity. Also, it needs to be in higher resolution. Tables 1 and 2, while comprehensive, could benefit from a summary discussion to highlight the main trends.

- Raw Data and Code: The authors have provided a link to the codebase and raw data.

Experimental design

- Research Questions: The research questions and objectives are relevant and well-defined. The claim that the proposed method is interpretable while maintaining performance is significant for medical AI.

- Methodology: The methodology for training Proto-Caps is described in detail. However:
* The hyperparameter choices (e.g., push-step interval, prototype count) lack adequate justification and appear tuned without clear rationale. It could be a good idea to have an ablation study.
* While the segmentation head is included in the architecture, its role and importance are underexplored. For instance, how does segmentation affect the attribute and target accuracy?
* The evaluation on two datasets is appreciated, but the choice of these datasets could have been justified better, especially when generalizing claims about interpretability. How do these datasets represent the problem that the authors are trying to solve?

- Reproducibility: The descriptions of datasets, loss functions, and architectures are comprehensive, but additional details on training times, hardware configurations, and optimization strategies (e.g., learning rate decay) would improve reproducibility.

Validity of the findings

- Interpretability Claims: The paper extensively evaluates interpretability, but it does not critically compare the interpretability of Proto-Caps with post-hoc explainable methods. Claims such as "Proto-Caps offers superior local explainability" need stronger empirical or user-study evidence.

- Performance Comparison: The comparative analysis with state-of-the-art methods is strong, but:
* The 3D results for the LIDC-IDRI dataset reveal slightly lower performance compared to non-explainable methods. This tradeoff should be discussed more explicitly.
* While attribute-based evaluations are included, they lack a thorough examination of why some attributes perform poorly (e.g., spiculation in 3D models).

- Statistical Soundness: The use of cross-validation and statistical reporting (mean and standard deviation) is commendable. However, some claims, such as the correlation between attributes and target predictions, could benefit from more robust statistical tests.

Additional comments

Compared to the MICCAI conference paper, the journal version has undergone several enhancements and expansions in both the scope and depth of analysis. While the conference paper introduced the Proto-Caps method and demonstrated its feasibility on one dataset, the journal version broadens the scope of evaluation, deepens the analysis of explainability and reliability, and explores various hyperparameters. It also incorporates user feedback considerations and experiments on an additional dataset and in 3D, which I think deserves a thumbs up.

Additional Comments:
- Lack of Critical Discussion: The manuscript does not sufficiently discuss the limitations of the proposed method, especially the potential challenges in real-world deployment (e.g., robustness to unseen data).
- Overreliance on Prototypes: While prototypes are claimed to enhance interpretability, their diversity and generalization capability are insufficiently analyzed. For example, how does the method handle rare or ambiguous cases?
- User Study Limitations: The user study is referenced but not elaborated upon. Detailed results and analysis of this study would strengthen claims about Proto-Caps' usability.
- Include more critical comparisons with post-hoc explainable methods and discuss the tradeoffs between intrinsic and post-hoc explainability.
- Provide additional results on model robustness (e.g., under distribution shifts or noisy labels) to demonstrate real-world applicability.
- Clarify the role and performance impact of the segmentation head in the architecture.
- Improve the accessibility of raw data and pre-processing scripts to facilitate reproducibility.
- Expand the discussion on how Proto-Caps can be extended to other domains or modalities.


Typos:
- Inconsistent use of American and British English (e.g., utilised, optimize, analyse)
- unterstandable, attibute-specific, an prediction

---

## Round 0.2 · accepted · Accept

Dear Authors,
Your paper has been revised. It has been accepted for publication in PEERJ Computer Science. Thank you for your fine contribution.

·

Basic reporting

No comment

Experimental design

No comment

Validity of the findings

No comment

Additional comments

Great job on the effort for this overhaul revision. My issues are all resolved with the revision. Thank you for the great work!